# Low-Intensity Continuous Ultrasound Therapies—A Systematic Review of Current State-of-the-Art and Future Perspectives

**DOI:** 10.3390/jcm10122698

**Published:** 2021-06-18

**Authors:** Sardar M. Z. Uddin, David E. Komatsu, Thomas Motyka, Stephanie Petterson

**Affiliations:** 1Department of Orthopaedics and Rehabilitation, Stony Brook University, Stony Brook, NY 11794, USA; David.Komatsu@stonybrookmedicine.edu; 2Department of Osteopathic Manipulative Medicine, Campbell University, Buies Creek, NC 27506, USA; motyka@cambelli.edu; 3The Orthopedic Foundation, Stamford, CT 06905, USA; spetterson@ofals.org

**Keywords:** ultrasound, low-intensity continuous ultrasound, sustained acoustic medicine, low-intensity pulsed ultrasound, soft tissue healing, regeneration, bone healing, sonophoresis, pain management, arthritis

## Abstract

Therapeutic ultrasound has been studied for over seven decades for different medical applications. The versatility of ultrasound applications are highly dependent on the frequency, intensity, duration, duty cycle, power, wavelength, and form. In this review article, we will focus on low-intensity continuous ultrasound (LICUS). LICUS has been well-studied for numerous clinical disorders, including tissue regeneration, pain management, neuromodulation, thrombosis, and cancer treatment. PubMed and Google Scholar databases were used to conduct a comprehensive review of all research studying the application of LICUS in pre-clinical and clinical studies. The review includes articles that specify intensity and duty cycle (continuous). Any studies that did not identify these parameters or used high-intensity and pulsed ultrasound were not included in the review. The literature review shows the vast implication of LICUS in many medical fields at the pre-clinical and clinical levels. Its applications depend on variables such as frequency, intensity, duration, and type of medical disorder. Overall, these studies show that LICUS has significant promise, but conflicting data remain regarding the parameters used, and further studies are required to fully realize the potential benefits of LICUS.

## 1. Introduction

The human body is overly sensitive to mechanical impetus [1,2,3,4,5,6]. Extensive research has shown that mechanical forces play an essential role in the growth, development, repair, and remodeling of organs and their component tissues by regulating cellular and molecular pathways [5,7,8,9,10]. The mechanotransductive forces cause stresses and strains in the extracellular matrix, resulting in confirmation changes at cellular and molecular levels. A wide range of data has shown that whole-body vibration can activate the regeneration of multiple organs [3,11,12,13,14].

Ultrasound (US) is a series of mechanical waves with a frequency >20 kHz [15,16]. When these cyclic waves propagate through a medium, they induce mechanical forces. Medical US applications vary depending on ultrasound parameters such as frequency, duty cycle, wavelength, energy, power, and intensity [17,18,19]. US parameters determine how the ultrasound waves are transmitted, attenuated, and reflected through the tissue. In most medical applications, reflected US waves are used for diagnostic imaging, while attenuated and transmitted waves are used as therapeutic agents. The intensity level (power/unit area) also plays an essential role in US. Over short intervals, intensities <0.1 W/cm^2^ are used for Doppler effects and intensities ranging between 0.1 and 1 W/cm^2^ are used for diagnostic imaging [17,19]. US intensities >10 W/cm^2^ generate a significant amount of heat. Thus, high-intensity focused US (HIFU) is mostly used for cancer and surgical applications [20,21,22,23]. In contrast, low-intensity US (20 to 1000 mW/cm^2^) produces relatively little heat over time, depending on duty cycle, frequency, wavelength, and treatment duration [16,24,25]. Low-intensity US can be further subdivided into pulsed and continuous US depending on the duty cycle (Figure 1). Pulsed US consists of ON/OFF cycles of US waves, while continuous US consists of continuous acoustic waves with no ON/OFF cycles. Numerous studies have examined the effectiveness of low-intensity pulsed US (LIPUS) and low-intensity continuous US (LICUS) as potential means of tissue regeneration and treatment for disorders such as pain, thrombosis, fracture healing, osteoporosis, and osteoarthritis [24,26,27,28,29,30,31,32,33].

US has multiple biological effects, including thermal, cavitation, microstreaming, nutrient exchange, and oxygenation [34,35,36,37]. The thermal effects of US are related to ultrasonic parameters and tissue density, with US impedance and attenuation determining the level of heat generated in the tissue and subsequent increases in vasodilation, oxygenation, and nutrient exchange [37]. Ultrasound’s properties of cavitation and microstreaming are attributed to the continuous cycle of compression and refraction of microbubbles within inertial fluids leading to localized stress and loosening of the matrix and cell membrane, thereby increasing the exchange of nutrients and promoting drug delivery [36]. Considering the multifactorial effects of US, it is an attractive therapeutic tool to target multiple degenerative, cancerous, and infectious diseases. LIPUS impact has been mainly attributed to its mechanical vibration due to minimum thermal effects at the tissue and cellular level. In contrast, LICUS stimulation has both mechanical and thermal effects at the tissue and cellular level. There are numerous comprehensive reviews that have described the effectiveness of LIPUS for various medical applications [24,38,39,40,41], but to our knowledge, there is no comprehensive review summarizing the therapeutic efficacy of LICUS. In the last few decades, there have been significant developments in LICUS applications. This work has led the FDA to approve two LICUS systems, namely: sam^®^ (ZetrOZ Systems, LLC, Trumbull, CT, USA) for soft tissue regeneration and pain management and UltraMIST^®^ (SANUWAVE Health, Inc.) for wound healing. Here, we will review pre-clinical and clinical studies conducted using LICUS to probe its therapeutic potential. This review focuses on the effectiveness of LICUS for tissue regeneration, the treatment of multiple disorders, and drug delivery (Figure 2).

## 2. Methods of How the Search Was Conducted for the Review Article: Materials and Methods

A stepwise search of Pubmed and Google Scholar databases was conducted. The primary search was conducted, using the keywords “low-intensity ultrasound" and “low-intensity therapeutic ultrasound”. Duplicates were excluded, and the secondary search was conducted for low-intensity continuous ultrasound (keywords: “low-intensity continuous ultrasound”); only English articles with keywords were reviewed. Studies describing pulsed low-intensity ultrasound were excluded. The literature review was conducted on 2 March 2021. To ensure integrity, only articles with well-defined ultrasound parameters were included in the review. All the articles missing essential parameters such as the duration of treatment, duty cycle, frequency, and intensity were excluded, as illustrated in the PRISMA flow chart (Figure 3).

## 3. Applications of Low-Intensity Continuous Ultrasound

### 3.1. LICUS Effects on Tissue Regeneration

Soft tissue injuries can be divided into open wounds and deep tissue injuries (stress, strains, sprains, and tears). De Lucas et al. reviewed the application of ultrasound therapy and the underlying activation of molecular and cellular pathways leading to cellular differentiation, proliferation, migration, matrix regeneration, and its biological effects in regenerative medicine [38]. The review extensively discussed LIPUS applications in clinical studies and respective regenerative outcomes in various tissues. Similarly, LICUS has been shown to be effective in treating both open wounds and deep tissue injuries [42,43,44]. Healing is a complex process initiating from the onset of the injury within minutes, leading to acute inflammation followed by angiogenesis, proliferation, tissue formation, and remodeling. Multiple synergistic cellular and molecular responses control the healing process. The initial acute inflammation process is vital to initiate the healing process [45,46]. The initial phase of inflammation attracts macrophages to localized damaged tissue to facilitate the breakdown of damaged tissue and regulate the inflammation process. The inflammation process is followed by angiogenesis, the migration and proliferation of neighboring cells to the damaged tissue, and matrix formation. LICUS enhances nitric oxide levels (NO), leading to increased vasodilation, blood flow, oxygenation, and an increase in the supply of nutrients [47]. The increased blood flow also facilitates macrophage migration into the injury site to regulate acute inflammation [48,49]. An in vitro study conducted by Altland et al. applied LICUS at 27 kHz at 0.25 W/cm^2^ for 10 min and found increases in the nitric oxide (NO) level by 102 ± 19% in human endothelial cells [47]. Sugitta et al. have shown increasing NO levels through the nitric oxide synthase (NOS)-dependent pathway with increasing LICUS intensity between 0.21 and 0.48 W/cm^2^, independent of thermic effects via stimulation of the abductor muscle in a rabbit model [49]. Similarly, LICUS promoted angiogenesis through regulating phosphoinositide 3 kinase, serine/threonine kinase, and the endothelial nitric oxide synthase (P13K-Akt-eNOS) pathway in human umbilical vein endothelial cells and a type 2 diabetic mouse peripheral ischemia model [50]. A study conducted by Huang et al. observed microtube formation on day four in human umbilical vein endothelial cell cultures after three stimulation sessions of 9 min per day with LICUS at 1 MHz, 0.3 W/cm^2^ intensity [50]. Karnes et al. reported improved muscle strength after a rat skeletal muscle injury following seven days of LICUS at 1 MHz, 0.5 W/cm^2^ for 5 min/day. The injured rats showed significant improvements in isometric tetanic force relative to the sham-treated group. LICUS treatment has also been shown to increase fibroblast proliferation, capillarization, and myofiber formation in a rat muscle injury model [51]. A comparative study between pulsed and continuous ultrasound by Vásquez et al. showed that both US forms increased the gastrocnemius muscle area significantly, with a more significant increase recorded in rats stimulated with LICUS at 3 MHz relative to 3 MHz, 50% duty cycle pulsed and sham control [52]. Furthermore, the application of LICUS increased the rate of differentiation of embryonic stem cells to cardiomyocytes relative to LIPUS [52]. 

The effects of US-induced regeneration are not limited to muscle tissue [53,54]. Whiting et al. reported improved chondrocyte proliferation in ex vivo cultures using 16- to 17-day-old fetal murine metatarsals following pulsed and continuous ultrasound exposure for four days [55]. The findings from the intermittent stimulation of continuous ultrasound at 5 MHz for 51 s twice per day for 10 days showed a significant increase in chondrocyte proliferation and matrix production [56]. LICUS stimulation also significantly promotes native-to-native cartilage integration in vitro. Sahu et al. simulated bovine cartilage and osteochondral explants at 5 MHz and 2 MHz for 20 min, four times/day for 28 days. At study completion, samples treated with LICUS displayed improved alignment and gap closures relative to untreated controls. Subsequent immunofluorescent analyses showed the presence of collagen II filling the native-to-native cartilage gap, and mechanical testing revealed a significant increase in the strength of LICUS-treated samples. Similarly, the enhancement of chondrocyte migration was demonstrated using a scratch assay [57]. Jackson et al. used an Achilles’ tendon puncture model to evaluate the recovery rate with and without LICUS (1.5 W/cm^2^, 4 min/day) stimulation at days 2, 4, 9, and 21. The effectiveness of LICUS was quantified by the formation of new collagen fibers and biomechanical stress. LICUS stimulation significantly increased tendon breaking strength, tensile strength, tensile stress, and energy absorption. Additional analysis of labeled proline in hydroxyproline showed a significant increase in collagen fibers following five days of LICUS treatment [48]. A similar study used a surgically immobilized Achilles tendon rabbit model that was stimulated with LICUS at 0.5 W/cm^2^ for 5 min/day for nine days. The authors reported significantly increased tensile strength, tensile stress, and energy absorption capacity [2]. Sparrow et al. confirmed the effectiveness of LICUS using a medial collateral ligament healing rabbit model [58]. After the transection, the rabbits were treated with 1 MHz, 0.3 W/cm^2^ for 10 min/day. The study reported significantly larger ligaments at six weeks concomitant with increases in tensile strength, tensile stiffness, and energy absorption [58]. 

In a clinical study, Best et al. recruited 20 subjects with elbow tendinopathy [53]. Subjects were treated with LICUS at 3 MHz, 0.132 W/cm^2^ (sam^®^, ZetrOZ) for 4 h per day for six weeks. After six weeks, subjects reported a 3.95 ± 2.15-point decrease in pain on an 11-point numeric rating scale (NRS) and a 2.83 ± 5.52-kg improvement in grip strength [53]. The recent series of case studies conducted by Draper et al. using LICUS (sam^®^, ZetrOZ) reported the effectiveness of LICUS in treating sports-related soft tissue injuries [59]. 

LICUS has been shown to be effective in treating chronic wounds and diabetic ulcers [42,60]. Studies conducted by Ennis et al. applied UltraMist^®^ therapy (Sanuwave Health, Inc.) using low-intensity continuous ultrasound at 40 kHz, 1.25 W/cm^2^ for up to 12 min, depending on wound size. In a clinical study (*n* = 29), Ennis et al. reported a 69% improved healing rate, as quantified by reduced area and volume of chronic wounds after 10 weeks of treatment [60]. Furthermore, Ennis et al. showed the efficacy of UltraMist^®^ in diabetic foot ulcers in 133 patients at 23 clinical sites. Patients were treated at 40 kHz and 1.25 W/cm^2^ from a 65-micron distance for 4 min with UltraMist therapy and showed a 40.7% healing rate over a 12-week follow-up period [42]. 

LIPUS is FDA approved for bone fracture healing of nonunions and has shown a 40% reduction in healing time for fresh fractures [16,61,62]. In contrast, there are no clinical studies reporting on the use of LICUS for bone healing. Yang et al. failed to identify significant bone formation using LICUS (125 mW/cm^2^) for 15 min/day for four weeks in a rat sciatic-neurotomy-induced bone loss model [6]. However, El-Bialy et al. studied the application of pulsed (20% duty cycle,1.5 MHz, 30 mW/cm^2^) and continuous (1.5 Mz, 30 mW/cm^2^) ultrasound for 20 min in mandibular osteodistraction in skeletally mature rabbits [63]. The authors reported significantly increased bone volume/tissue volume over four weeks [63]. Further studies are required to better understand the effects of LICUS on bone formation in the short- and long-term.

### 3.2. LICUS Role in Pain Management

Chronic pain management is a significant clinical challenge [64,65,66,67]. Acute pain is a complex process, but in the simplest form, it has three phases: (1) stimulus conversion to chemical signal, (2) chemical impetus and synaptic events changed into electrical events in the neurons and (3) electrical events transduced into chemical events at the synapse [68]. The continuous and unregulated activation of any component of this pathway can lead to chronic pain. Numerous therapeutic drugs and rehabilitation methods have been designed to interrupt the translation of acute pain to chronic pain. Medications can be effective, but the long-term utilization of systemic medications leads to significant adverse effects on various physiological systems [64,67]. Thus, there is a need to develop new therapeutics to target pain-specific phases. Multiple studies have used US as a targeted, non-invasive method for pain management as a stand-alone or adjunct therapy in conjunction with other traditional treatments such as physical therapy, exercise, and RICE (rest, ice, compression, and exercise) [30,69,70,71,72,73,74,75,76,77,78,79,80,81,82,83,84,85].

LICUS can alleviate musculoskeletal pain by increasing local temperature, vasodilation, and increasing metabolism. Several studies have shown the effectiveness of LICUS as a stand-alone or adjunct therapy to manage soft tissue pain. Muftic et al. reported an improved visual analog score (VAS) for pain. The study included 68 patients treated with 10 LICUS sessions, at either 0.4 W/cm^2^ for 8 min or 0.8 W/cm^2^ for 4 min. Both men and women responded positively to LICUS, with women responding more positively to LICUS treatments [86]. Patterson et al. used the sam^®^ device for long-duration stimulation of LICUS at 3 MHZ, 132 mW/cm^2^ intensity for latent trigger points in the upper trapezius, shoulder, and neck to relieve tissue stiffness, alleviate pain, and increase quality of life [79]. After four weeks of treatment, the global rate of change assessment was significantly higher in the stimulated group (2.84 ± 2.21 to 0.46 ± 2.08 points (*p* < 0.001)) and a 1.28-point improvement in pain numerical rate scale scores after four weeks of treatment. A randomized, placebo-controlled clinical trial (*n* = 27) conducted by Yildirim et al. reported a reduction of 30% in VAS pain during activity in LICUS-treated patients after 10 seasons of LICUS treatment at 1 MHz and 1.5 W/cm^2^, compared to the placebo group, which was treated with blinded inactive US [85]. Srbely and Dickey reported on the antinociceptive effectiveness of LICUS by attenuating the sensitivity and pain response of myofascial trigger points [87]. The study reported 22 patients in the test group stimulated directly over the trigger point locations using LICUS at 1 MHz, with an intensity of 1 W/cm^2^ for five minutes. Pain pressure threshold (PPT) measurements were taken at the trigger point location immediately following LICUS treatment. LICUS increased the mean PPT by 44.1% compared to a 1.4% increase in the control group. LICUS treatments also significantly reduced sensitivity at the trigger point. The authors suggested that LICUS-induced upregulation of nitric oxide synthase combined with deep tissue heating reduces trigger point sensitivity. LICUS has also been shown to be effective in elbow and Achilles tendinopathy [87]. Best et al. confirmed the effectiveness of LICUS in managing the pain and dysfunction associated with elbow tendinopathy at the lateral epicondyle. A total of 20 subjects receiving 4 h per day of LICUS using the sam^®^ device for six weeks reported a 3.94 ± 2.15 (*p* = 0.0002) point decrease in pain as well as a 2.83 ± 5.52 kg increase in grip strength (*p* = 0.04) [53]. These effects appear to be restricted to LICUS as LIPUS has been reported to have little or no effect on chronic patellar tendinopathy or muscle soreness [88,89]. 

Osteoarthritis (OA) is a progressive, degenerative, and inflammatory disease. Multiple studies have shown the efficacy of LIPUS on preventing the progression of arthritis, but recent data suggest the greater effectiveness of LICUS at slowing the progression of arthritis [39,72,90,91,92,93,94,95,96]. Chung et al. studied the application of LICUS in the early stages of arthritis on neutrophil activation in a rat model [72]. The administration of 1 MHz ultrasound for 10 min daily for one to four days showed a significant increase in apoptosis of neutrophils [72]. Park et al. examined the use of LICUS in combination with hyaluronan (HA) injections in a rabbit knee OA model. The rabbits’ knees were treated with LICUS at 1 MHz, 400 mW/cm^2^ for 10 minutes. The stimulation increased the intake of HA and its half-life. This combination therapy of LICUS and HA injections dramatically increased the volume of proteoglycans and prostaglandin E2 in the synovial fluid, and reduced collagen II and matrix metalloproteinase-3 in the OA-induced joint [94,95]. 

Clinical trials conducted by Tasciooglu et al. showed significant improvement in the Western Ontario and McMaster Universities Index (WOMAC) after a short-term, two-week study of 5-min sessions a day per week of LICUS stimulation in patients with knee OA [97]. Similarly, Draper et al. examined the efficacy of LICUS in a double-blinded study in 90 patients with knee OA [98]. Patients were distributed into treated (*n* = 55) and placebo (*n* = 35) groups. After six weeks of 4 h daily treatment with LICUS at 3 MHz, 0.132 W/cm^2^, and 1.3 W per day, patients treated with active devices reported a mean 1.96 numerical rating scale (NRS) pain reduction relative to a 0.85 NRS reduction in the placebo group (treated with non-active ultrasound device). Significant improvements were also reported for secondary functional measures, including mobility, stiffness, and pain. The authors suggest that LICUS is a potential non-invasive therapy for knee OA [98]. Similarly, the application of LIPUS has also shown encouraging trends in clinical and preclinical studies. A meta-analysis conducted by Zhou et al. compared five clinical studies and concluded that LIPUS treatment has favorable effects on reducing osteoarthritis-associated pain and increases ambulation speed [28]. Another meta-analysis comparing LIPUS and LICUS reported that LICUS significantly reduced osteoarthritis-associated pain but did not improve mobility, while LIPUS reduced pain as well as increased mobility. Studies show that both continuous and pulsed ultrasound have no adverse effects and more evidence is required to establish which modality has the most clinical promise [93].

### 3.3. Regulation of Neuromodulation

Recent data have shown that US stimulation can actively modulate central and peripheral nervous systems and play a significant role in neuron activity, suppression, and proliferation [24,29,99]. US stimulation of the nervous system can have short-term and long-term effects. In the short-term, US stimulation directly activates neurons in the hippocampus and controls the translation of the neuronal signal through synaptic sodium and calcium channels [99,100,101,102,103,104,105]. In the long-term, US stimulation modulates levels of the extracellular neurotransmitters—serotonin, dopamine, and ƴ-aminobutyric acid—potentially enabling the treatment of epilepsy and other CNS-associated disorders. A recent study by Xin et al. reported that LIPUS and LICUS decreased local field potentials, phase-amplitude coupling strength, and the interval between seizures in an epileptic mouse model [103]. Studies conducted by Liu et al. and Zhao et al. have demonstrated long-term neuroprotective effects of US stimulation in neurodegenerative disorders, such as Parkinson’s and Alzheimer’s [104,105,106]. Researchers have used both LIPUS and LICUS to evaluate the effectiveness of US in neuromodulation. King et al. reported in a comparative study that LICUS is more effective than LIPUS in activating motor neurons in mice, and its effects on motor neurons are correlated to LICUS intensity and duration [107]. On the contrary, Kim et al. reported that focused LIPUS is superior to LICUS in activating motor neurons [108]. These studies show the effectiveness of US in CNS neuromodulation, but it remains unclear whether LIPUS or LICUS is more beneficial [104].

US stimulation of the peripheral nervous system (PNS) shows similar trends. Downs et al. conducted a comparative study by applying multiple duty cycles of US (15, 35, 5, 90, 100%, 3.75 MHz) for 4.5- to 10-millisecond stimulations in a mouse sciatic nerve crush model. Histological and electromyography analyses were used to assess nerve recovery. The data showed that low-intensity US at 100% (LICUS) showed the best recovery relative to other groups [109]. Ni et al. confirmed the effectiveness of LICUS using a rat sciatic nerve crush model. LICUS treatment at 1 MHz, 0.2 W/cm^2^ for 1 min/day for four weeks significantly improved sciatic nerve functional index (SFI) and muscle recovery, as well as increased the levels of brain-derived neurotrophic factor at weeks 3 and 4 relative to non-treated controls [110]. Despite encouraging results in vivo experiments, clinical studies by Leonid and Gavrilov reported LIPUS activating more skin electroreceptors relative to LICUS-stimulated skin receptors [111]. Kim et al. conducted a systemic review of preclinical and clinical studies of low-intensity transcranial ultrasound and concluded that LIPUS could positively modulate superficial and deep cerebral tissue and control cognitive or motor behavior [112]. Sanguinetti et al. have reported that LIPUS has inhibitory modulatory effects and can be used to control mood [113]. Lee et al. demonstrated that the application of LICUS did not result in a significant increase in sensation, but stimulation at lower frequencies (350 KHz) did significantly increase the vibrotactile and nociception response in human fingertips [114]. No significant adverse short- or long-term effects have been reported for either CNS or PNS studies in human or animal studies. While there are limited data available regarding the efficacy of LICUS for neuromodulation, thus far, studies have shown that the effectiveness of US on PNS neuromodulation is highly dependent on specific US parameters. 

The exact mechanism of the US effect on PNS neuromodulation is not well-known, but to trigger action potentials, US can potentially trigger one of the four following mechanisms: (1) the activation of capacitive currents by membrane displacement; (2) induction of pores in the lipid bilayer, known as sonoporation; (3) activation of mechanosensitive channels; and (4) transmission of the acoustic waves along axons. These mechanisms can activate neuromodulation via the activation of individual mechanisms or a combination of multiple mechanisms [104].

### 3.4. LICUS Effectiveness in Thrombosis

Several studies have shown the effectiveness of US for articular and deep vein thrombolysis [32,33,115,116,117,118,119]. Both high- and low-intensity US has been shown to be effective in the thrombolysis and fibrinolysis of blot clots. High-intensity US utilizes the thermal and cavitation aspect of US to ablate the clots, while low intensity activates urokinase and streptokinase to dissolve blood clots over time. In vitro studies conducted by Francis et al. reported the dissolution of plasma clots using LICUS at 1 MHz with up to 8 W/cm^2^ intensity [116,120]. No mechanical disintegration of the plasma clot was observed, and increased activity was seen for urokinase and streptokinase [116,120]. Similar data have been reported by other groups with lower intensity US [115,117,119,121]. 

At higher intensity, thrombolysis occurs due to an increase in localized temperature and cavitation. In vivo data confirm the phenomenon observed in in vitro experiments. Riggs et al. showed ablation of femoral artery thrombosis in a rabbit model with 1 MHz and 2 MHz LICUS stimulation, while increasing the intensity to 6.3 W/cm^2^ did not show similar results [122]. Suchkova et al. showed enhanced streptokinase activity in a rabbit femoral artery thrombosis model with minimal heating effects and histological integrity when stimulated at 40 kHz, 0.75 W/cm^2^ using LICUS. The application of LICUS at 40 kHz, 0.8 W/cm^2^ showed a significant decrease in cerebral infarct in a rat middle cerebral stroke model [123]. The intensity of US can directly target clots using a catheter to minimize vessel wall damage and heating effects. Trubestein et al. applied high-intensity US at 26.5 kHz frequency delivered through a catheter to mechanically disturb clots with minimal cell damage in dog iliac vessels [121]. Similar results were seen by Nedelmanna et al., who used LICUS with a US array of 20, 40, and 60 kHz at 0.2 W/cm^2^ to test the ablation of a blood clot in vitro settings. The study reported the highest ablation of clots at the lower frequency (20 kHz), suggesting that lower frequencies are more suitable for thrombolysis [117,124]. Further studies are required to assess the effectiveness of LICUS and optimize parameters. A clinical study conducted by Chen et al. showed the effectiveness of LIPUS as a combined therapy with alteplase in intravenous thrombosis for vascular recanalization in acute ischemic stroke [125]. Additionally, Aguiar et al. demonstrated the efficacy of LIPUS in reducing microvascular obstruction and improving myocardial dynamics in patients with ST-segment elevation [126].

### 3.5. Sonophoresis and Drug Delivery

Targeted, non-invasive, and sustained drug delivery remains an unresolved clinical challenge [127,128]. The cavitation and acoustic streaming ability of LICUS make it a promising tool for drug delivery [34,129,130]. Skin is a compact structure of three layers—epidermis, dermis, and hypodermis. The dermal layer is highly vascularized. Drugs need to pass through the epidermal layer to reach the dermis and be effective. The epidermal layer (stratum corneum) resembles a brick (corneocytes) and mortar (lipid matrix) structure. Various non-invasive methods have been applied to enhance the penetration of drugs through the epidermal layer, such as chemical enhancers, iontophoresis, electroporation, microneedles, laser ablation, pressure waves, photochemical waves, and radiofrequency waves [130,131,132]. Unfortunately, all these technologies have had little success in providing effective transdermal drug delivery. 

US offers a viable alternative to current technologies due to its diverse bioeffects on skin [34,130,131,133]. Sonophoresis is driven by the collapsing or imploding of unstable high-energy bubbles generating localized shockwaves and cavitation. The formation of microjets and shockwaves loosens the surrounding matrix and increases local temperature, allowing the drug to penetrate through the tightly compact stratum corneum. Mitragotri et al. reported that increasing the frequency has an inverse relation with sonophoresis efficacy. Higher US frequencies reduce the rate of formation of microbubbles and cavitation and lower-frequency US has achieved a higher rate of absorption [130,131]. 

US intensity is dependent on the medium threshold and its acoustic decoupling [127]. Studies have shown increased transdermal drug delivery between 0.1 and 0.3 W/cm^2^ at frequencies of 20 to 100 kHz and 0.1 to 10 W/cm^2^ at frequencies of 1 to 3 MHz [130,131]. Miyazaki et al. evaluated the transdermal drug delivery of ointment in a rat at different intensities (0.25, 0.5, 0.75, 1 W/cm^2^) at 1 MHz, and reported that 0.75 W/cm^2^ was the most effective intensity when used for a 10-min duration [134].

Similarly, pulse duration plays an integral part in the efficacy of sonophoresis. LICUS increases tissue temperature; thus, it is essential to retain lower frequency and intensity to reduce tissue damage while enhancing tissue permeability [35,37,135]. Multiple studies have shown the effectiveness of LIPUS in drug delivery across the skin, blood–brain barrier, and in tumorigenesis [136,137,138]. Cagnie et al. reported a 10-fold increase in ketoprofen levels in synovial tissue after being exposed to LICUS at 1 MHz, 1.5 W/cm^2^ for 5 min relative to the topical application of ketoprofen [139]. In addition, an ex vivo study conducted by Aldwaikat et al. evaluated the penetration of diclofenac sodium. The authors concluded that 20 kHz, 20% amplitude continuous ultrasound increased drug delivery by 546% relative to the control group who received a commercially available topical ointment, EpiDerm^®^. Furthermore, the authors reported that the change was independent of changes in US amplitude. [140] Langer et al. used a long-duration LICUS approach to study the penetration of salicylic acid in hydrogel transdermal drug delivery in vitro. Their SonoBandage system applied US at 10%, 50%, and 100% duty cycle at frequencies of 175 kHz and 3 MHz for 1 and 4 h. The study found a significant increase in drug penetration using LICUS for 4 h relative to non-treated controls [133]. Similarly, Masterson et al. investigated diclofenac sodium penetration through a hydrogel stack using long-duration LICUS at 3 MHz, 0.132 W/cm^2^ for 4 h. They reported a 3.8-fold higher drug penetration through a hydrogel stack mimicking the ex vivo transdermal model [141]. In a recent multicenter clinical study, Madzia et al. reported treating 32 patients with moderate to severe knee OA with sam^®^ (LICUS at 3 MHz, 0.132 W/cm^2^, 1.3 W) for 4 h/day for one week in conjunction with sodium diclofenac patches [142]. After seven days of treatment, patients showed a 2.06- to 2.96-point decrease in pain on the NRS pain scale and 351- to 510-point functional improvement on the WOMAC scale. To further amplify the effects of LICUS, Schoellhammer et al. applied a dual-frequency approach to enhance drug delivery. The study used a combination of 20, 40, and 60 kHz at 1 MHz and 3 MHz intensity for 2 and 6 min. A significant increase in acoustic cavitation was seen with the 20 and 40 kHz frequencies. Furthermore, the study reported an increase in glucose and insulin sonophoretic delivery levels ex vivo using porcine skin [143]. In a follow-up in vivo study, Schoellhammer et al. showed the greater effectiveness of combined 20 kHz and 1 MHz relative to just 20 kHz sonophoresis of dextran, along with a favorable safety profile for dual-frequency US in vivo [144]. Yin et al. reported similar data using 20 and 800 kHz frequencies, enhancing the delivery of sinomenine hydrochloride into porcine skin [145]. 

The application of LICUS is not limited to sonophoresis through the skin. Zderic et al. used LICUS at a frequency of 880 kHz and intensities ranging between 0.19 to 0.56 W/cm^2^ for 5 min to deliver hydrophilic dye in rabbit corneas. More than 10 times the amount of dye was delivered at 0.56 W/cm^2^, which disappeared within 90 min with reported damage to epithelium cells [146,147]. Kine-Schoder et al. applied LICUS at multiple frequencies (400 kHz, 600 kHz, 800 kHz, and 1 MHz) at 1 W/cm^2^ intensity to evaluate the penetration through the nail with the possibility of improving the treatment of onychomycosis. The study reported that 800 kHz and 1 MHz resulted in significantly higher penetration of dye through the nail relative to the sham control [146]. Pong et al. demonstrated the effectiveness of LICUS at 20 kHz, 0.13 W/cm^2^ using an in vitro wound healing model using phospholipids liposomes [148]. These applications of LICUS for sonophoresis show its potential as a tool for targeted drug delivery.

### 3.6. Cancer Treatment

Studies have used low-intensity US as sonodynamic therapy, including US-mediated chemotherapy, US-mediated gene delivery, and anti-vascular US therapy [25,27,149,150]. These therapies depend on the ability of the US to induce cellular cavitation and increase temperature. Sonodynamic therapy induces cavitation and sonosensitizers produce free radicals, killing cancer cells with minimal damage to surrounding tissues [27,151]. Sonodynamic therapy has shown the most effective results between 1.0 and 2.0 MHz at an intensity of 0.5 to 3.0 W/cm^2^ [130,131]. In vitro studies by Zeng et al. and Sawai et al. have shown the effectiveness of LIPUS in treating cancer by inhibiting the ERK1/2 and AKT pathways [152,153]. Yu et al. used LICUS at 0.24 MHz in chemo-resistant ovarian tumor cells to show the effectiveness of LICUS-driven chemotherapy in in vitro and in vivo models [154]. Jin et al. demonstrated that sonodynamic therapy at 1 MHz, 0.51 W/cm^2^ for 10 min reduced cancer cell growth by 77% in a squamous cell carcinoma C3H/HeN mouse model and significantly increased the survival rate [155]. Barati et al. used dual LICUS (1 MHz and 150 kHz) for 30 days to treat pre-clinical breast adenocarcinoma in Balb/c mice. The study reported a delay in tumor growth rate relative to sham-treated groups [156,157]. Other studies have also shown LICUS-induced carcinogenic effects in in vitro and in vivo models. Numerous investigators have used different sonosensitizers to optimize cancer cell apoptosis by inducing cavitation, free radicals, and local thermal effects through cell membrane destruction, mitochondrial swelling, and chromatin condensation. US with chemotherapeutic-loaded microbubbles, micelles, and liposomes has been used to enhance the delivery of chemotherapeutic agents at the tumor site and induce cancer cell apoptosis. Inertial cavitation drives microbubbles into tumor cells to improve chemo-agent delivery at the tumor site [130]. Hayashi et al. have shown the enhanced induction of cell death and apoptosis of human malignant glioma cells by combined treatment with the photosensitizer Photofrin and LICUS at 0.3 W/cm^2^ for 3, 15, and 30 min [158]. Yang et al. studied the synergetic effects of microbubbles and LICUS-induced chemotherapy at 1.1 MHz, 2.0 to 4.0 W/cm^2^ and showed a significant increase in doxorubicin uptake by cancer cells relative to monotherapy [159]. In vitro and in vivo studies have shown the increased efficacy of tumor cell apoptosis using US-enhanced chemotherapy. Staples et al. applied LICUS at 20 kHz, 1 W/cm^2^ with micelles infused with doxorubicin and reported a higher concentration of chemotherapeutic agent within tumor cells within 30 min post-stimulation. The authors suggested that the increased vasculature of the tumor, along with cavitation, increases the delivery of chemotherapeutic agents [160]. A recent study by Loria et al. reported that LICUS at 1 MHz, 120 mW/cm^2^ for 15 min enhanced the delivery of human ferritin-based nanoparticles loaded with a chemotherapeutic agent and resulted in selective apoptosis of colon adenocarcinoma (HT29), colorectal carcinoma (HCT116), and sarcoma (SW982, SW872 cell lines) [161]. LICUS-induced cancer apoptosis is not limited to chemotherapeutic agents. Li et al. reported increased gene delivery of a recombinant plasmid into the MCF-7 breast cancer cells. The genes were transfected using LICUS at 2 MHz, 0.75 W/cm^2^ for 45 s. The study reported a 21.92 ± 3.64% increase in gene uptake by cancer cells without affecting the non-cancer cells. The study demonstrated that the LICUS-derived gene delivery system is a safe method to target cancer cells [162]. Wood et al. used an anti-vascular approach by employing high-frequency LICUS at 2.4 W/cm^2^ to increase the temperature of tumor tissue, leading to localized capillary rupture and reduced tumor volume, which increased the survival rate in mice with implanted melanomas [163]. Tardoski et al. evaluated the effects of LICUS in a bone metastasis mouse model; LICUS at 2.9 MHz for 30 min produced a hyperthermic impact on the tumor and increased the uptake of zoledronate, triggering bisphosphonate anticancer activity [164]. There are multiple therapies targeting cancer, yet cancer treatment remains an ongoing challenge. LICUS provides a potential tool to directly target cancer cells to reduce cancer progression. The application of US in tumorigenesis is currently limited to in vitro and preclinical studies.

## 4. Future Perspectives

Despite encouraging data regarding US treatment for multiple disorders over several decades, its application has been primarily limited to pre-clinical studies. There are various reasons for the failure of the transition of pre-clinical studies to clinical adoption. The first and most important hurdle is the optimization of US parameters, as the efficacy of US is dependent on multiple parameters, including frequency, duty cycle, wavelength, energy, power, and intensity. The optimization process is further reliant on the biology of the disorder. For example, parameters that work for the regeneration of soft tissue healing may not be the best parameters for bone healing, considering the nature of anatomy, physiology, and morphology. The density and type of tissue target also play a vital role in considering energy loss as acoustic waves penetrate through multiple tissue layers. Therefore, proper animal and cell models must be used to study the effectiveness of US. All aspects of proper use of in vitro cell lines, 3D cultures, animal models, and surgical approaches play a role in transitioning pre-clinical studies to clinical studies. The experimental design, sample size, and demographics will also play a vital role in the success of any clinical treatments. 

In addition to optimizing US parameters, the size and technical expertise required to use US equipment has limited the clinical adoption of therapeutic US. Currently, therapeutic US is mostly used for rehabilitation, which requires frequent patient visits to rehabilitation centers as well as trained staff to deliver wound-healing technologies such as UltraMist^®^ [42,60]. Current advances in technology have allowed US devices to miniaturize, leading to FDA approval, such as EXOGEN^®^ and sustained acoustic medicine (sam^®^) [62,81,165,166]. EXOGEN (Smith & Nephew) uses LIPUS at a 20% pulse rate, 1.5 MHz, 30 W/cm^2^ for 20 min per day for bone healing (nonunion) [62]. sam^®^ (ZetrOZ Systems LLC, Trumbull, CT, USA) uses LICUS at 3 MHz, 0.132 W/cm^2^, and 1.3 W for 4 h per day for soft tissue pain management and tissue regeneration [81,166]. The development of these systems allows for the in-home utilization of US, but their efficacy remains limited by patient compliance. 

Most US research has focused on LIPUS, but recent data from the last two decades show encouraging results using LICUS in different disorders, as reviewed above. LICUS has already been approved for clinical applications in wound healing, pain management, and soft-tissue regeneration. Additionally, both clinical and pre-clinical data show the significant potential of LICUS in neuromodulation, cancer treatment, and thrombolysis. Further research is required to optimize the parameters to make additional transitions from pre-clinical studies to clinical studies possible. Studies have shown that the effectiveness of LICUS is highly dependent on a range of US parameters [131]. Therefore, it is imperative to consider the biology of the targeted tissue and how acoustic properties change through the tissue. The transition of LICUS to clinical studies is highly dependent on the choice of correct pre-clinical models and rigorous statistical analyses. Thus, it is essential to consider all the aspects of the pre-clinical models and its relevance to target clinical implications. 

Importantly, there are only a few head-to-head studies of LICUS and LIPUS; thus, there is a need for studies directly comparing LICUS and LIPUS by only altering the duty cycle and keeping all other US parameters constant to conclusively determine which is better for a given indication [108,167]. LIPUS biological effects are highly dependent on mechanotransductive pathways, while LICUS can activate both thermal and mechanotransductive pathways, so there are likely preferential applications for each modality. In conclusion, this article systemically reviewed LICUS as a promising non-invasive, inexpensive treatment for multiple disorders. It showed that LICUS is being used successfully to treat pain and soft tissue injuries and has tremendous promise for new clinical applications from oncology to thrombosis treatment and drug delivery.

## Figures and Tables

**Figure 1 jcm-10-02698-f001:**
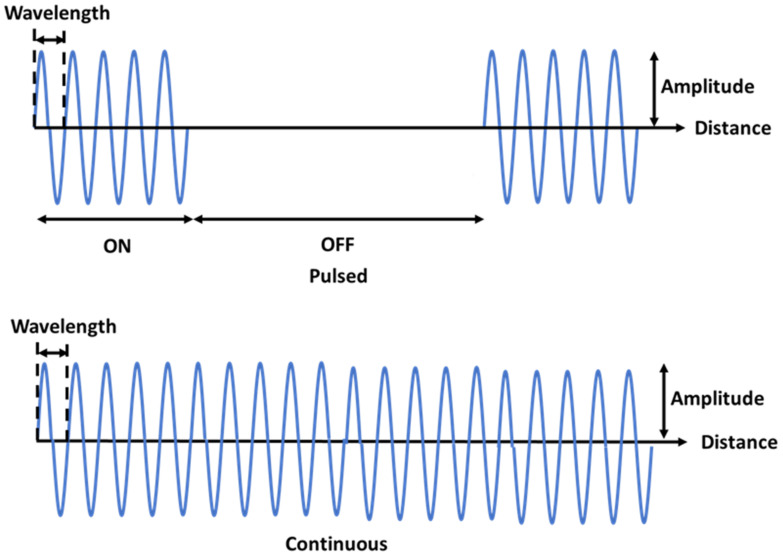
Illustration of the fundamental difference between pulsed and continuous ultrasound waveforms. Top: pulsed US waveform showing ON/OFF cycles. Bottom: a continuous US waveform is a continuous wave with no ON/OFF cycles.

**Figure 2 jcm-10-02698-f002:**
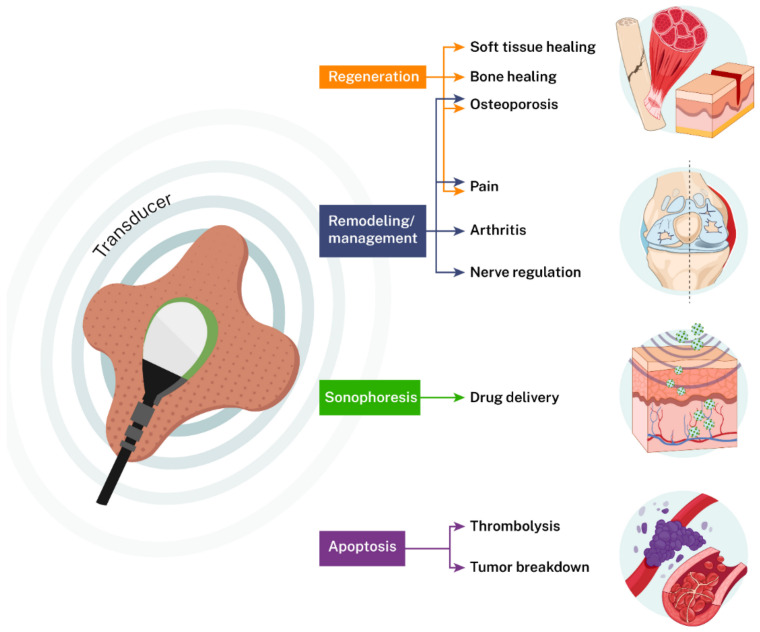
Applications of LICUS in medicine. LICUS can be used to regenerate soft and hard tissues, pain management, sonophoresis, thrombosis, and apoptosis.

**Figure 3 jcm-10-02698-f003:**
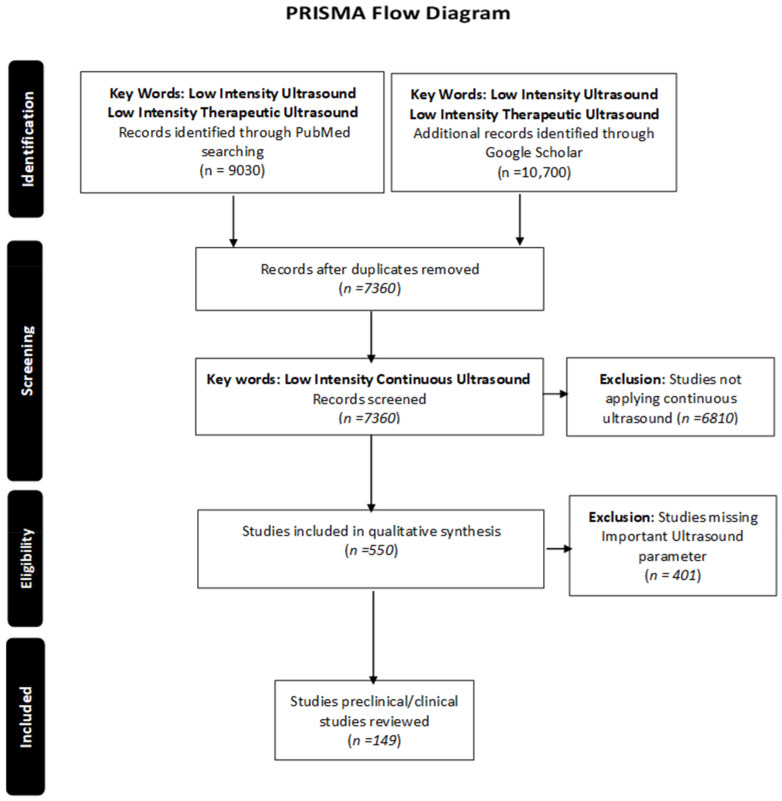
PRISMA flow chart: describing the methodology of article selection for review.

## Data Availability

Not applicable.

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
