# Peer review of "Low-Intensity Continuous Ultrasound Therapies—A Systematic Review of Current State-of-the-Art and Future Perspectives"

_jcm, 2021, doi:10.3390/jcm10122698_

Round 1
Reviewer 1 Report
The article submitted by Sardar Uddin and co-authors reports a detailed review on Low Intensity Continuous Ultrasound (LICUS) for therapeutic applications.
The topic is of interest. Low Intensity Ultrasound is a totally non-invasive tool able to promote beneficial outcomes in different medical domains, even if treatment optimization and a deeper understanding still need to be achieved.
However, in my opinion, the paper should better clarify which could be the benefits of using a continuous stimulation with respect to pulsed ones. The non-lethal mechanical effects within the “low pressure regime” of LIPUS are widely explored nowadays and the advantage of a pulsed stimulation with respect to a continuous stimulation is quite clear in the community of ultrasound researchers and users: namely the minimization of thermal effects.
The authors should clarify this important point and highlighting which could be the advantages of LICUS with respect of LIPUS. In addition, the works in which LICUS is compared to LIPUS - by changing only the duty cycle and keeping constant all the other parameters (frequency, pressure/intensity, duration) - should be emphasized more than the studies in which only the continuous stimulation was explored.
I also suggest to add one or more summary tables recapping the US parameters, the model used, the bioeffect, … of the most important LICUS works.
Other minor points:
- Line 38. The upper limit of US is not 10 MHz (also for medical US). In general, better define US as a mechanical wave with a freq > 20 kHz;
- Figure 1. If in the x axis is “Time” the distance between two consecutive peaks is the “Period” and not the “Wavelength” which is a distance.
- Figure 3. The number of articles of “studies included in qualitative synthesis” should be = 7365-6801 and not 168.
Author Response
Response to Reviewer 1 Comments
Reviewer 1:
Point 1: "However, in my opinion, the paper should better clarify which could be the benefits of using a continuous stimulation with respect to pulsed ones. The non-lethal mechanical effects within the "low-pressure regime" of LIPUS are widely explored nowadays, and the advantage of a pulsed stimulation with respect to a continuous stimulation is quite clear in the community of ultrasound researchers and users: namely the minimization of thermal effects"
Response 1: The authors appreciate the reviewer's comment and understand the importance of comparing continuous ultrasound with pulsed ultrasound and its well-reported benefits. The objective of this systemic review article was to comprehensively review recent advances in Low-Intensity Continuous Ultrasound, as numerous review articles have already focused on the applications of Low-Intensity Pulsed Ultrasound. However, in response to this suggestion, we have added references to recent LIPUS publications review articles and studies in each section of our revised manuscript.
Point 2: "The authors should clarify this important point and highlighting which could be the advantages of LICUS with respect of LIPUS. In addition, the works in which LICUS is compared to LIPUS - by changing only the duty cycle and keeping constant all the other parameters (frequency, pressure/intensity, duration) - should be emphasized more than the studies in which only the continuous stimulation was explored.
I also suggest to add one or more summary tables recapping the US parameters, the model used, the bioeffect, … of the most important LICUS works."
Response 2: The authors agree with the importance of discussing studies in which only the duty cycle is altered while keeping all other parameters constant (frequency, pressure/intensity/duration). Unfortunately, few studies have compared the effects of changes in the duty cycle while keeping other parameters consistent. We agree that these studies are essential to conclusively determine if pulsed or continuous ultrasound is more beneficial for any specific clinical modality and have added such a recommendation in the future perspectives section.
The authors appreciate the suggestion of adding one or multiple tables to study the change in bioeffects in changing parameters of LICUS in different clinical conditions. However, given the broad variety of indications, parameters, models, and effects, we feel that this would not be any more concise than the existing text.
“Other minor points:
- Line 38. The upper limit of US is not 10 MHz (also for medical US). In general, better define US as a mechanical wave with a freq > 20 kHz;
- Figure 1. If in the x axis is “Time” the distance between two consecutive peaks is the “Period” and not the “Wavelength” which is a distance.
- Figure 3. We have updated the figure.
Minor Changes: Thanks for pointing out these minor issues. All the changes have been made as recommended by the reviewer.

Reviewer 2 Report
I would like to congratulate with the Authors. Extensive research in databases was conducted, and extensive review of LICUS applications has been done. I would suggest to complete (or remove) Table 1 that is not comprehensive of all FDA-approved devices (nor all applications have been included into the Table).
Author Response
Point 1:"I would like to congratulate with the Authors. Extensive research in databases was conducted, and extensive review of LICUS applications has been done. I would suggest to complete (or remove) Table 1 that is not comprehensive of all FDA-approved devices (nor all applications have been included into the Table)."
Response 1: The authors appreciate the kind and encouraging words from the reviewer. Complying with the reviewer's comment, we have deleted Table 1.

Round 2
Reviewer 1 Report
In Figure 1, I suggest to change "Distance" with "Time" and "Wavelength" with "Period". It is more intuitive for understanding the difference between continuos and pulsed waves.